# Genetic Factors of Idiopathic Gigantomastia: Clinical Implications of Aromatase and Progesterone Receptor Polymorphisms

**DOI:** 10.3390/jcm11030642

**Published:** 2022-01-27

**Authors:** Anna Kasielska-Trojan, Michał Pietrusiński, Magdalena Bugaj-Tobiasz, Jerzy Strużyna, Maciej Borowiec, Bogusław Antoszewski

**Affiliations:** 1Plastic, Reconstructive and Aesthetic Surgery Clinic, Institute of Surgery, Medical University of Lodz, Kopcinskiego 22, 90-153 Lodz, Poland; boguslaw.antoszewski@umed.lodz.pl; 2Department of Clinical Genetics, Medical University of Lodz, 92-213 Lodz, Poland; michal.pitrusinski@umed.lodz.pl (M.P.); maciej.borowiec@umed.lodz.pl (M.B.); 3Eastern Centre of the Burns Treatment and Reconstructive Surgery in Leczna, 21-010 Leczna, Poland; magdalena.bugaj@tlen.pl; 4Department of Plastic, Reconstructive and Burns Surgery, Medical University of Lublin, 21-010 Leczna, Poland; jerzy.struzyna@umlub.pl

**Keywords:** gigantomastia, aromatase, progesterone, polymorphism

## Abstract

The role of estrogen, progesterone, their receptors and aromatase in the development of the breast is well documented. In this study we examined the association of genetic variants of progesterone receptor (PGR) and aromatase (CYP19A1) genes with gigantomastia risk. We conducted a case-control study among 124 women: 60 with gigantomastia and 64 controls. We examined the single nucleotide polymorphisms (SNPs) for CYP19A1 (rs749292 and rs7172156) and PGR (rs1042838). Our results showed that allele G in rs749292 (CYP19A1) increased the risk of gigantomastia, but not significantly (*p* = 0.09). There is a correlation between rs1042838 (PGR) and waist-to-hip ratio (WHR) in women with gigantomastia-AC genotype correlates with lower WHR and CC with higher WHR. There were no correlations between the onset of gigantomastia, the age of menarche and the length of the menstrual cycle, and rs1042838, rs749292 and rs7172156. We did not find differences in the SNP of PGR (rs1042838) between women with gigantomastia and controls. However, our findings showed more frequent G allele in CYP19A1 (rs749292) in women with gigantomastia.

## 1. Introduction

Breast hypertrophy, also known as gigantomastia, is a rare condition characterized by excessive breast growth. Physical and psychological symptoms are criteria for the diagnosis, rather than the volume of excessive breast tissue [1,2,3]. Due to the fact that the definitions of gigantomastia are not standardized, there are no exact figures for the incidence of this condition [4]. Juvenile virginal enlargement of the breast is the most common type of gigantomastia. It can be seen in late childhood and is often present between the age of 11 and 14 years [1,5]. The underlying etiology of this condition is unclear. However, it has been speculated as due to an abnormal response or sensitivity of breast tissue to hormones during puberty [6,7]. Previous studies, which examined hormonal levels did not reveal any imbalances and no increased estrogen receptor concentration was detected [8,9,10,11]. In our previous study we found that there is no primary overexpression of estrogen and progesterone receptors in women with gigantomastia and that they had, at some stage of development, higher circulating levels of androgens, which in breasts are converted to estrogens due to aromatase [12,13]. However, there are still many unverified hypotheses concerning breast hypertrophy etiology, e.g., abnormal sensitivity of estrogen (ER) or progesterone receptors (PGR) or aromatase overexpression. Verification of these hypotheses could allow for consideration of the role of different pharmacotherapies in juvenile breast hypertrophy (e.g., tamoxifen, aromatase inhibitors). On the other hand, it may also help to establish if women with gigantomastia are at higher risk of breast cancer and other sex-hormone related cancers [14,15].

The aim of this study was to identify genetic risk factors for breast hypertrophy among genes for aromatase (CYP19A1) and progesterone receptor (PGR), examining their single nucleotide polymorphisms (SNPs) (CYP19A1: rs749292 and rs7172156 and PGR: rs1042838). A rationale for choosing these polymorphisms was based on previous studies suggesting a possible role of aromatase in the development of this condition and on the basis of the literature review [11,12,16]. Comparison of gene polymorphisms between women with gigantomastia and control women may help to explain the etiology of this condition and may be helpful in predicting the risk of sex-hormone related cancers in women with gigantomastia.

## 2. Materials and Methods

### 2.1. Study Population

The study involved 124 participants (Caucasian women), including 60 patients who underwent breast reduction due to juvenile/idiopathic gigantomastia in two Plastic Surgery Centers located in different regions of Poland (central and east) (Table 1). The average age of the studied individuals was 39.6 years (SD = 11.12 years). All patients qualified for surgical treatment had undergone endocrine examinations and a detailed medical interview had also been taken to exclude any possible reason for breast enlargement. They had undergone breast ultrasonography and/or mammography examination. All surgical procedures were performed because of therapeutic indications due to clinical conditions caused by excessive breast weight (mastalgia, neck and back pain, headaches, trophic lesions of the breast skin with ulceration and infection, limited ability to exercise) and the resection weight was minimum 1000 g per breast. Exclusion criteria included any hormonal disturbances or treatment (current or past, excluding contraceptives), obesity (BMI > 30 kg/m^2^), pregnancy-related gigantomastia, any abnormalities in breast, history of breast malignancy, only aesthetic reasons for breast reduction, and reductions less than 1000 g per breast. Removed tissues were routinely sent for histological examination.

Among Medical University students (mean age 25 years, SD = 4.11 years), 64 women were recruited as control. All women declared their breast cup sizes from A to D and had no history of breast cancer.

### 2.2. Measurements

The study consisted of a clinical questionnaire concerning demographic data, history of breast hypertrophy, family history of gigantomastia and a genetic investigation.

#### Genetic Study

DNA was isolated from buccal cells using a commercial kit for isolation of genomic DNA, Sherlock AX, A&A Biotechnology, Gdynia, Poland according to manufacturer’s instructions. The resulting nucleic acid was suspended in TE buffer and stored at −20 °C for further analysis. The SNPs located at the genes for aromatase (CYP19A1, rs749292 and rs7172156) and progesterone receptor (PGR, rs1042838) were evaluated. The G>A (CYP19A1, rs7172156, the minor allele frequency (MAF) A–0.400216), G>A (CYP19A1, rs749292, MAF-A = 0.441183), and the C>A (PGR, rs1042838, MAF-A = 0.163099) SNPs were determined using TaqMan^®^ SNP Genotyping Assays (Thermo Fisher Scientific Inc., Waltham, MA, USA). Reactions were carried out using the AriaMx Real-Time PCR thermocycler (Agilent, Santa Clara, CA, USA) in 96-well plates, in a volume of 20 μL, according to manufacturer’s instructions.

### 2.3. Statistical Analysis

Logistic regression was used to evaluate the relationship between the genetic variant and the risk of gigantomastia Four genetic models were considered: dominant—GG vs. AA + AG (or CC vs. AA + AC), recessive—AA vs. AG + GG (or AA vs. AC + CC), over-dominant—AG vs. AA + GG (or AC vs. AA + CC), multiplicative (allele-based). To avoid the increase of type I error due to multiple comparisons, we chose the best model before calculating the OR for each model [18]. We also examined MAF for the controls and compared it with MAF for the European population (The ALFA project, from dbGaP). To evaluate correlations of SNPs with clinical variables we used: χ^2^ for the history of breast hypertrophy in mother, grandmother, ANOVA for breast volume and waist-to-hip ratio (WHR), and Kruskall-Wallis test for the age of the onset of gigantomastia, the age of menarche and the length of the menstrual cycle. In case of significant ANOVA results, we used Scheffe’s test to ascertain which pairs of means are significant.

A significance level of *p* < 0.05 was accepted. All statistical analyses were performed using the STATISTICA package (v13, StatSoft, Cracow, Poland).

The protocol for the study was approved by the local ethics committee of the Medical University of Lodz, RNN/06/18/KE.

## 3. Results

As shown in Table 2 and Figure 1 the multiplicative model appeared to be the best. Gigantomastia was not observed to be significantly associated with rs7172156, rs749292 and rs1042838 in the multiplicative model. However, allele G of aromatase rs749292 (CYP19A1) increased the risk of gigantomastia (the probability for the homozygous GG genotype was 60%), but not significantly (*p* = 0.09). Also allele A of aromatase rs7172156 (CYP19A1) increased the risk of gigantomastia (the probability 62% for the homozygous AA genotype, *p* = 0.15) (Table 2 and Table 3). Allele A in rs7172156 CYP19A1 appeared to be significantly less frequent in our control sample (23.44%) than in the general European population (40.02%) (*p* = 0.0028), but its frequency in women with gigantomastia was similar to the populational (*p* = 0.269). There were no differences in the minor alleles’ frequency between the groups in rs1042838 PGR (gigantomastia 17.5% vs. controls 14.06% vs. EU population 16.31%) and in rs749292 CYP19A1(gigantomastia 48.33% vs. controls 59.38% vs. EU population 44.12%).

There were no differences between the analyzed gene variants and family history of gigantomastia (a positive history of gigantomastia in: mothers—χ^2^ = 1.248, *p* = 0.54; grandmothers—χ^2^ = 2.95, *p* = 0.23). Anthropometric variables did not correlate with CYP19A1 SNPs. None of the gene variants correlated with breast volume (F = 0.959; *p* = 0.39), but we found a correlation between PGR rs1042838 and WHR. Scheffe test indicated that the difference is related to differences between genotypes AC (lower WHR) and CC (higher WHR) (Table 4). There were no correlations between the onset of gigantomastia, the age of menarche and the cycle length, and rs1042838, rs749292 and rs7172156.

## 4. Discussion

The etiology of juvenile/idiopathic gigantomastia is unknown and examination of this issue seems to be important as it could help to answer questions about the possible risk of breast cancer and other sex-hormone dependent cancers in these women as well as provide a basis for pharmacotherapy of this disease. Kusano et al. (2006) found that for lean women larger breast size was associated with a higher risk of breast cancer and Eriksson et al. (2012) showed that two single nucleotide polymorphisms, which had previously been associated with breast cancer risk, are also associated with breast size [19,20]. There are, however, no studies concerning this issue in women with gigantomastia and, to our knowledge, this study is the first to examine the possible relationship between genetic variants and this condition. In this study we focused on analyzing aromatase gene polymorphisms and progesterone receptor gene polymorphism in women with gigantomastia. Our previous studies showed a possible role of aromatase in the development of this condition, while specific polymorphisms were chosen on the basis of a literature review [11,12,16]. Although, we did not find any significant differences between a control group and women with this condition, we observed that allele G in aromatase rs749292 (CYP19A1) (major allele in the European population) was more frequent in women with gigantomastia. We also found a correlation between PGR SNP (rs1042838) and WHR in women with gigantomastia-genotype CC correlated with higher WHR.

There are two functional PGR polymorphisms: PROGINS and +331G/A. PROGINS is a haplotype of three genetic variations in complete linkage disequilibrium: Val660Leu substitution (rs1042838), silent-His770His (rs1042839) and a HS-1/PV Alu insertion. It has a frequency of 0.12–0.16 among Caucasians and reduces the activity of progesterone receptors [21,22]. There are studies reporting the association of genetic variation in PGR with breast cancer. The minor allele frequency of PGR rs1042838 was significantly higher in breast cancer patients compared to controls. Patients carrying rs1042838 A allele had higher risk of breast cancer and the allele was associated with Her2 status. [23,24]. PGR rs1042838 also appeared to be a risk factor for endometrial cancer (allele A increases the risk) [25]. In our analysis this PGR SNP (C/A) was not associated with gigantomastia, but allele C correlated with higher WHR and genotype AC was associated with lower WHR in women with gigantomastia. Our previous study showed that women with breast hypertrophy have higher WHR than control women [12]. Present results may indicate that women with gigantomastia and lower WHR (indicative of genotype AC) may be at higher risk of endometrial cancer. This observation, however, is preliminary and should be verified on a larger sample of participants.

We also analyzed two clinically significant polymorphisms of the aromatase gene (rs7172156 and rs749292 in CYP19A1). Their inclusion in our analysis was based on the study which reported that the minor alleles of rs749292 were positively and the minor alleles of rs7172156 were inversely associated with daily 17β-estradiol. Moreover, the rs749292 minor alleles were inversely associated with absolute mammographic density-lean women with rs749292 minor alleles had lower risk for high absolute mammographic density, while lean women with rs7172156 minor homozygous genotype had risk for high absolute mammographic density [16]. In their meta-analysis, Goodman et al. (2008) found that the A allele of rs749292 was positively associated with ovarian cancer risk in a codominant model for all races combined [26]. Similarly, Setiawan et al. (2009) showed association between the A allele of rs749292 or rs727479 in CYP19A1 and the increased risk of endometrial cancer [27]. Our study showed a tendency for lower frequency of allele A of rs749292 in women with gigantomastia. This may suggest that estrogen levels may not be involved directly in pathogenesis of gigantomastia. Moreover, in our previous study we showed that women with gigantomastia have higher WHR, which also suggests that, if breast hypertrophy develops due to estrogenic stimulation, this is rather local estrogenic over-stimulation, e.g., due to local aromatase overexpression [12]. This may also suggest that women with gigantomastia do not present higher risk of sex hormone dependent cancers. On the other hand, authors from Iran found the association between CYP19A1 (rs749292) (also for CYP2C8 (rs1058930), CYP1B1 (rs1056836)) genes’ polymorphisms and increased risk of breast cancer in women in Mazandaran province. G allele of rs749292 increased the risk of breast cancer [28]. Our analysis did not show differences between alleles’ frequency of aromatase gene in women with gigantomastia and controls; however, in the case of CYP19A1 (rs749292), we found more frequent allele G in women with gigantomastia. As Flote et al. (2014) reported, lean women who had rs749292 minor alleles had lower risk for high absolute and high percent mammographic density compared with major homozygous genotype [16]. Such “low risk genotype” was less frequent in lean women with gigantomastia, examined in our study. This may suggest that further studies on mammographic characteristics of women with gigantomastia should be performed.

There are some limitations to the study. First, the study cohort is small, resulting in low statistical power for detection of associations. However, the limited sample size in this study has some justifications. Most of all, gigantomastia is not a common condition (no statistical data on the prevalence is available for the Polish population) and, considering its diagnosis is based on physical symptoms, it creates a rather heterogenous group of disorders. To overcome this problem, we imposed rigorous inclusion criteria—we included only women with juvenile or idiopathic gigantomastia (cases with gestational gigantomastia, hormonal disturbances, and breast pathological masses were excluded). An additional criterion that limited the number of participants was resection weight of at least one kilogram per breast. These criteria helped to recruit a rather homogenous group of women with gigantomastia. Applying such standards resulted in a small number of participants so we included patients from another center where such surgeries are often performed and where an accurate recruitment for the project was secured. To further increase the number of patients meeting these criteria, more centers should be included and a multicenter project could be established. Secondly, we included only white women from two regions of the country, which does not allow for generalization. The results are preliminary and provide rationale for further studies examining PGR and CYP19 polymorphisms in women with gigantomastia, e.g., also including intron 4 (TTTA)n repeat and TCT deletion/insertion polymorphisms [29].

Another methodological aspect that should be commented on is the rationale for choosing female students as a control group, as their age is lower than in the case of women with gigantomastia. However, age-matching did not seem to be relevant as age does not influence gene polymorphism and, with high probability, the controls did not include women who will develop juvenile or idiopathic gigantomastia, as the mean age of gigantomastia onset in our participants was 14.6 years, much lower than the controls’ mean age.

To conclude, we did not find significant differences in the single nucleotide polymorphisms (SNPs) of genes for aromatase (CYP19A1, rs749292 and rs7172156) and progesterone receptor (PGR, rs1042838) between women with gigantomastia and controls. However, we observed that allele G in rs749292 (CYP19A1) increased the risk of gigantomastia. We found a correlation between A allele of rs1042838 in PGR (correlated in previous studies with endometrial cancer) and lower WHR in women with gigantomastia, which may suggest that women with this condition and lower WHR may be at higher risk of endometrial cancer. Further multicenter studies on larger samples are needed to establish the etiology of gigantomastia and the risk of sex-hormone dependent cancers in these women.

## Figures and Tables

**Figure 1 jcm-11-00642-f001:**
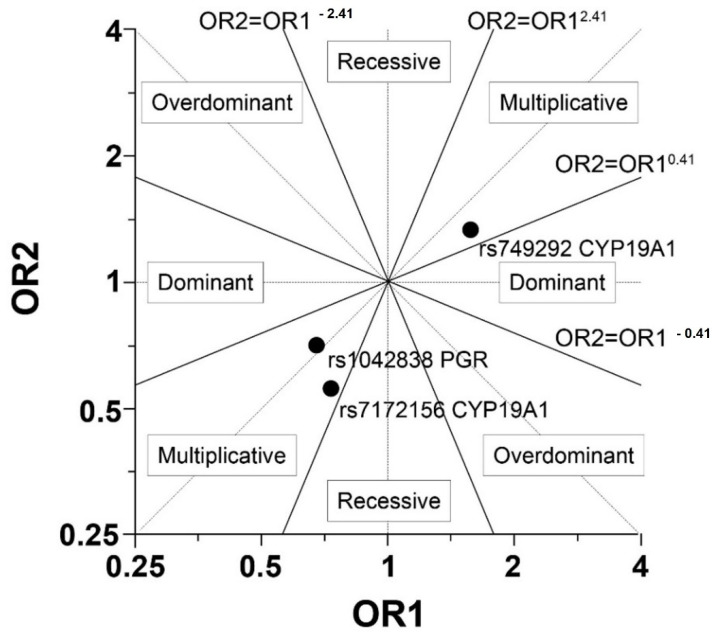
Graphical explanation of the genetic model choice (OR—Odds Ratio, OR1 = odd AG/odd AA and OR2 = odd GG/odd AG (or OR1 = odd AC/odd AA and OR2 = odd CC/odd AC).

**Table 1 jcm-11-00642-t001:** Study participants’ characteristics.

Characteristic	Women with Gigantomastia *n* = 60 Mean ± SD
Age (years)	39.6 ± 11.12
Age of gigantomastia onset (years)	14.6 ± 3.5
Breast volume (right/left) (cm^3^) *	897/871
Age of menarche (years)	12.8 ± 1.5
Length of the cycle (days)	28.7 ± 2
BMI (kg/m^2^)	25.8 ± 3.1
WHR **	0.89 ± 0.07
Family history for gigantomastia (yes)	37

* measured with BrestIdea Volume Estimator [17], data for 40 patients, ** data for 43 patients, BMI—Body Mass Index, WHR—Waist-to-Hip Ratio, SD—Standard Deviation.

**Table 2 jcm-11-00642-t002:** Logistic regression analysis of the association between single nucleotide polymorphisms (genotype frequency distribution) of aromatase and progesterone receptor genes in women with gigantomastia and controls. Choice of the best model in four model strategy.

		Original Data				Second Step				
		Controls *n* = 64 (%)		Gigantomastia *n* = 60 (%)		OR	(95% CI)	*p*-Value	(OR1; OR2) *	Model
rs7172156 CYP19A1	AA	6	(9.38)	9	(15)	1.29	(0.38; 4.36)	0.685	(0.78; 0.64)	Mu
	AG	18	(28.13)	21	(35)	1.00 (ref)				
	GG	40	(62.5)	30	(50)	0.64	(0.29; 1.42)	0.274		
rs749292 CYP19A1	AA	23	(35.94)	14	(23.33)	0.61	(0.26; 1.42)	0.246	(1.64; 1.45)	Mu
	AG	30	(46.88)	30	(50)	1.00 (ref)				
	GG	11	(17.19)	16	(26.67)	1.45	(0.57; 3.68)	0.426		
rs1042838 PGR	AA	2	(3.13)	3	(5)	1.40	(0.20; 9.86)	0.734	(0.71; 0.82)	Mu
	AC	14	(21.88)	15	(25)	1.00 (ref)				
	CC	48	(75)	42	(70)	0.82	(0.35; 1.91)	0.637		

* OR1 = odd AG/odd AA and OR2 = odd GG/odd AG (or OR1 = odd AC/odd AA and OR2 = odd CC/odd AC), Mu—multiplicative model, OR—Odds Ratio, CI—Confidence Interval, ref—OR for the reference genotype = 1.0.

**Table 3 jcm-11-00642-t003:** Logistic regression for the multiplicative model for single nucleotide polymorphisms of aromatase and progesterone receptor genes in women with gigantomastia and controls.

Multiplicative Model SNP		Logistic Regression		
		OR	(95% CI)	*p*-Value
rs7172156 CYP19A1	AA/AG/GG	0.69	(0.41; 1.15)	0.154
rs749292 CYP19A1	AA/AG/GG	1.55	(0.93; 2.59)	0.09
rs1042838 PGR	AA/AC/CC	0.96	(0.73; 1.26)	0.771

OR—Odds Ratio, CI—Confidence Interval.

**Table 4 jcm-11-00642-t004:** Correlation between anthropometric features (breast volume and WHR) and genes’ polymorphisms.

		Mean Breast Volume *			WHR **	
SNP Variant	*n*	Mean ± SD	*p*-Value ^	*n*	Mean ± SD	*p*-Value ^
rs7172156_*CYP19A1*						
AA	7	791.21 ± 349.95	0.392	8	0.88 ± 0.09	0.716
AG	13	955.04 ± 169.82		13	0.91 ± 0.04	
GG	20	870.90 ± 272.71		22	0.89 ± 0.08	
rs1042838_*PGR*						
AA	3	109.17 ± 378.80	0.248	3	0.90 ± 0.06	0.028
AC	10	822.15 ± 267.21		10	0.84 ± 0.06	
CC	27	882.33 ± 240.45		30	0.91 ± 0.07	
rs749292_*CYP19A1*						
AA	11	858.64 ± 237.12	0.778	12	0.90 ± 0.07	0.879
AG	17	918.88 ± 267.22		18	0.89 ± 0.07	
GG	12	858.83 ± 283.34		13	0.89 ± 0.07	

^—ANOVA, * measured with BrestIdea Volume Estimator [17], data for 40 patients, ** data for 43 patients, WHR—waist-to-hip ratio, SD—standard deviation, SNP—single nucleotide polymorphisms.

## Data Availability

The data presented in this study are available on request from the corresponding author.

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
