# Peer review of "Genetic Factors of Idiopathic Gigantomastia: Clinical Implications of Aromatase and Progesterone Receptor Polymorphisms"

_jcm, 2022, doi:10.3390/jcm11030642_

Round 1

Reviewer 1 Report

In this paper the Authors has analysed genetic variants of PGR and CYP19A1 genes according to gigantomastia risk. For this purpose, 60 patients with gigantomastia and 64 participants were examined as a control group. 

  1. Approval of local ethical review Committee must be mentioned in Method section.
  2. The paper needs language editing.
  3. Abbreviations should be described the first time they are mentioned. E.g. WHR in line 20.

Reviewer 2 Report

Overview and general comments

In this study Kasielska-Trojan et al assessed the associations of three SNPs and clinical variables with gigantomastia risk. In their study they analysed data from 60 gigantomastia cases and 64 matched controls, all women of European ancestry recruited in two regions from Poland. The main findings were that there was some evidence to suggest CYP19A1 rs749292 was associated with gigantomastia, and that there were differences in WHR by PGR SNP genotypes.

I have several concerns about this study, which I have detailed as “major” and “minor” concerns. My biggest concern is the small sample size analysed and the resultant lack of power to detect associations if they truly exist. I understand that the condition under investigation is a rare condition, hence the recruitment of cases is a limiting factor. The authors should make more effort to discuss the sample size limitation and suggest ways in which this could be addressed by future research.

The statistical methods used to analyse the data may also not have been optimal. I feel that a regression-based approach (logistic regression) may be more appropriate than the methods selected by the authors.

Major concerns

Section 2.1: Could the authors justify/discuss the appropriateness of the control population to the case population. There appears to be a large disparity between the average age in these populations (case mean = 39.6y, control mean = 25.0y) and the variation in age was much larger in cases (SD = 11.12y) compared with controls (SD = 4.11y).

Section 2.1: Have the authors conducted any power or sample size calculations pre/post analysis? I feel that the sample size of 124 women (~60 case-control pairs in 1:1 ratio) would be greatly underpowered to detect associations, except for extraordinarily large effect sizes.
                Indeed, I did a quick sample size calculation using the freely available Quanto software, assuming approximate parameters from the PGR SNP (control frequency = 0.14, per-allele OR = 1.25) and a 1:1 matched case-control study. Assuming type I error rate of 0.05, the required sample size to achieve 80% power was ~1200 case-control pairs, and to achieve 90% power was ~1600 case-control pairs. The current sample size has ~10% power to detect associations.
                The authors do mention the issue of small sample size in the discussion (lines 220-221). Whilst I understand the condition being analysed is rare, I feel that a more comprehensive discussion of this study limitation should be addressed in the limitations section of the discussion. The authors should discuss what could be done to improve the sample size and power to detect associations.

Section 2.2: The rationale for analysing these SNPs is given in the discussion (lines 166-168) – whereas it should be outlined earlier in the article to orientate the reader as to why these SNPs/genes are being considered for their association with gigantomastia. If the authors are interested in these genes (CYP19A1 and PGR), then several other SNPs in these genes or SNPs not in these genes but regulating them may be functional/causal for gigantomastia. Why were further SNPs in/around these genes not analysed?

Section 2.3: I feel a more appropriate approach to the statistical analyses would be to follow a regression modelling approach (logistic regression for case-control data). This would enable readers to gauge estimated effect sizes (odds ratios). Whilst it is possible for readers to calculate these themselves from the case/control genotype counts, the authors should present the estimated effect sizes. I’m not certain why the authors are assessing the associations of the clinical variables with the SNPs. A strength of the regression-based approach is that one can adjust for (potential) confounding factors (for example, the clinical factors).

Section 2.3: The authors conduct many statistical tests (3 SNPs with 4 genetic model types, more clinical risk factors), hence I feel that they should be considering some sort of multiple testing correction to determine statistical significance. Whilst a Bonferroni correction may be too conservative (especially since each of these tests is not strictly ‘independent’ as the same SNPs are being considered multiple times under a different genetic model), there are other multiple testing corrections.

There is not a table describing the study participants in the results. This would be informative to readers.

Were study participants (systematically) tested for the Polish BRCA1 founder mutations? What impact might it have on the findings of this study if any participants were BRCA1 mutation carriers?

Minor concerns

Lines 28-29: The authors state that gigantomastia is a rare condition. Are there any published estimates of the prevalence/incidence of this condition that the authors could provide in the introduction?

Line 58: The authors refer to “different regions of the country”. Which country? I assume Poland, but this should be made clear here.

Section 2.1: The authors should state the ancestries of the women in this study. This is later included in the discussion, but this detail should be stated earlier in the methods.

Section 2.3: The Yates correction may be too conservative. Given the cell counts a Fisher Exact test may be more appropriate here, although as mentioned in the major concerns, a regression modelling approach may be more appropriate than the contingency table analyses presented by the authors.

Lines 94-95: The authors state that they compared the allele frequencies from controls against a reference population (ALFA project). There is no justification/reasoning for this. I assume it is to determine whether their samples align with what is expected in a European population. Could the authors explain why they are doing this?

Section 2.3: What statistical software was used to conduct the analyses?

Lines 103-104: Usage of the phrase “marginally associated” (meaning P-value just above 0.05) should be avoided. The authors should state in the methods what their assumed P-value threshold was.

Line 115: The authors perform a Scheffe test, although this was not mentioned in the methods section. They should describe what this test is being used for in the methods.

Other concerns

Line 97: The authors introduce the acronym “WHR” without mentioning what this stands for. I think it is “waist-hip ratio”, but in the first usage it should be spelled out.

Line 102 and lines 135-136: Use of the word “variants” here is confusing, as I would typically expect use of this word to be analogous to “SNP” i.e. meaning “genetic variation”. Perhaps a better choice of word here would be “model”.

Table 1: There is no need to show the chi-sq statistics. Presenting the P-values is sufficient. Showing an effect size, such as an odds ratio (mentioned previously), would be more useful for readers.

Table 2: No need to show the F-statistics, again P-value is enough here.

Tables 1-2: No need to report P-values to 4 decimal places. Usually two/three decimal places is enough.

Line 179: “Minor allele frequency” should read “The minor allele frequency”.

Round 2

Reviewer 2 Report

Most of my concerns have now been addressed by the authors in their revision. I still have some issues about the presentation of results in the tables. More extensive footnotes would help to understand the tables more readily, in the present form it is not immediately clear what the authors are showing and takes some time to fully understand what data are trying to be conveyed.

Lines 271-272: I think the wording could be better here, perhaps something along the lines of: “First, the study cohort is small, resulting in low statistical power to detect associations.”

Table 1 (and subsequent tables) – all abbreviations should be explained in the table footnotes. For example, “SD” in Table 1.

Table 2 is not entirely clear. It may help to have a single column for the OR, stating the OR for the reference genotype is 1.0, then have the genotype specific OR on the row for that genotype. It would also be informative to show confidence intervals and P-values in the table.

Table 3: I think there is too much information here, some of which is not needed. The t-statistic, probability of gigantomastia and power don’t need to be presented.

Table 4: Is “p” in this table a P-value or the correlation? I think this table needs a footnote to explain the table further. This is a general comment really and applies to all tables.

Author Response

Dear Editor and Reviewers,

Thank you for your interest in our manuscript entitled " Genetic factors of idiopathic gigantomastia: clinical implications of aromatase and progesterone receptor polymorphisms” and for your approval for the responses to your major comments. In this revision we addressed all your minor comments (in the text changes are marked in green colour (previous changes – in red)). We hope that this revision meets with your approval.

Sincerely,

The Authors

Responses to Reviewers’ comments:

  • Lines 271-272: I think the wording could be better here, perhaps something along the lines of: “First, the study cohort is small, resulting in low statistical power to detect associations.”

The sentence was changed as suggested.

  • Table 1 (and subsequent tables) – all abbreviations should be explained in the table footnotes. For example, “SD” in Table 1.

Done.

  • Table 2 is not entirely clear. It may help to have a single column for the OR, stating the OR for the reference genotype is 1.0, then have the genotype specific OR on the row for that genotype. It would also be informative to show confidence intervals and P-values in the table.

Table 2 was revised according to your suggestions (ORs, CIs and p were added).

  • Table 3: I think there is too much information here, some of which is not needed. The t-statistic, probability of gigantomastia and power don’t need to be presented.

Table 2 was revised according to your suggestions.

  • Table 4: Is “p” in this table a P-value or the correlation? I think this table needs a footnote to explain the table further. This is a general comment really and applies to all tables.

The tables were further explained (for table 4 “p” applies to ANOVA test)